



# **Aerosol concentrations variability over China:**

# **two distinct leading modes**

Juan Feng[1], Jianlei Zhu[2,*], Jianping Li[3], and Hong Liao[4]

*1.   College of Global Change and Earth System Science, Beijing Normal University, Beijing,*
*China*
*2.   Foreign Economic Cooperation Office, Ministry of Ecology and Environment, Beijing, China*
*3.   Key Laboratory of Physical Oceanography–Institute for Advanced Ocean Studies, Ocean*
*University of China and Qingdao National Laboratory for Marine Science and Technology,*
*Qingdao 266003, China*
*4.   School of Environmental Science and Engineering, Nanjing University of Information Science*
*& Technology, Nanjing, China*
***Corresponding author*:**
*Dr. Jianlei Zhu*
*Foreign Economic Cooperation Office, Ministry of Ecology and*
*Environment, Beijing, China*
Email: zhu.jianlei@fecomee.org.cn



## Abstract

Understanding the variability in aerosol concentrations (AC) over China is a scientific
challenge and is of practical importance. The present study explored the month-to-
month variability in AC over China based on simulations of an atmospheric chemical
transport model with a fixed emissions level. The month-to-month variability in AC
over China is dominated by two principal modes: the first leading mono-pole mode and
the second meridional dipole mode. The mono-pole mode mainly indicates enhanced
AC over eastern China, and the dipole mode displays a south-north out-of-phase pattern.
The two leading modes are associated with different climatic systems. The mono-pole
mode relates to the 3-month leading El Niño-South Oscillation (ENSO), while the
dipole mode connects with the simultaneous variation in the North Atlantic Oscillation
(NAO) or the Northern Hemisphere Annular Mode (NAM). The associated anomalous
dynamic and thermal impacts of the two climatic variabilities are examined to explain
their contributions to the formation of the two modes. For the mono-pole mode, the
preceding ENSO is associated with anomalous convergence, decreased planetary
boundary layer height (PBLH), and negative temperature anomalies, which are
unfavorable for emissions. For the dipole mode, the positive NAO is accompanied by
opposite anomalies in the convergence, PBLH, and temperature over southern and
northern China, paralleling the spatial formation of the mode. This result suggests that
the variations originating from the tropical Pacific and extratropical atmospheric
systems contribute to the dominant variabilities of AC over China.



## 1. Introduction

Aerosol particles are the primary pollutants in the atmosphere and play significant roles in influencing human health, environmental pollution, and regional and global climate (IPCC, 2013). The variation in aerosols shows considerable impacts on the climate via its direct and indirect effects by altering the radiation forcing and microphysical effects (e.g., Thompson, 1995; Zhang et al., 2011; Huang et al., 2006), indicating the important influences on the regional and global climate. For instance, it is noted that the 'cooling pool' in eastern-central China during the period 1960-1990 is partially attributed to increased aerosol concentrations (AC; Li et al., 2016), and that aerosols may exert influences on precipitation changes in both global and regional scales, as well as on monsoon systems (Rosenfeld et al., 2007; Cowan and Cai, 2011; Huang et al., 2014; Jiang et al., 2016; Lou et al., 2018). Thus, a better understanding of the AC variation is of significance for both scientific and practical efforts.

Meanwhile, the distribution and accumulation of aerosols are sensitive to meteorological conditions. The variations in the meteorological factors, e.g., precipitation, wind, temperature, planetary boundary layer height (PBLH), atmospheric stability, and humidity, could impact the AC by modulating the aerosol transport, deposition, and dilution processes (Aw and Kleeman, 2003; Lin and McElroy, 2010; Liao et al., 2015; Yang et al., 2017). The anomalies in meteorological conditions are attributed to the synoptic weather and climate systems. For the synoptic weather scale, Guo et al. (2014) indicated that stagnate weather conditions contribute to the periodic cycle of particulate matter events during boreal winter in Beijing. And the increase in



relative humidity (Han et al., 2014) and decrease in the PBLH (Quan et al., 2014; Yang
et al., 2015) would lead to an increase in the aerosols, thus contributing to the haze
events during winter 2012 in northern China.

Meanwhile, the variations in the large-scale climatic systems, such as Pacific

Decadal Oscillation (PDO), El Niño-South Oscillation (ENSO), East Asian summer
and winter monsoon (EASM & EAWM), and North Atlantic Oscillation (NAO) show
considerable effects in impacting the regional AC in both the seasonal and the
interannual timescales. For example, researchers found that low values of AC are
observed in Taiwan accompanied with the onset of the EASM (Chen and Yang, 2008).
During the mature phase of the moderate La Niña event 2000/01, an anomalous south-
negative-north-positive AC dipole pattern is seen over eastern China (Feng et al., 2017).
The interannual variations in the EASM exhibit significant effects in impacting the
summertime AC over China, i.e., high-level AC would be observed over eastern China
along with a weaker EASM (Zhu et al., 2012; Lou et al., 2016; Mao et al., 2017). A
similar situation is observed between the EAWM and AC over eastern China but during
boreal winter, showing that a weaker EAWM relates to a high level of AC over China
(Jeong et al., 2017). Zhao et al. (2016) have indicated that the decadal regime shift of
the PDO showed significant role in impacting the decadal variations of boreal winter
aerosols over eastern China. Feng et al. (2019) have reported the important influences
of simultaneous ENSO and preceding autumn NAO signals on the winter AC over
China by case study.



The above discussions highlight the effect of climate background in impacting the
AC over China across different seasons, including signals from both the tropical and
the extratropical, and originating from both the atmosphere and the ocean. However,
the relative roles of climate systems are still unknown because there are strong
interactions among the systems. For example, during the decaying summer of a warm
ENSO event, a weaker EASM is expected to be observed (Wu et al., 2002), and the
occurrence of a cold ENSO event during its mature phase is favorable for a stronger
EAWM (Wang et al., 2008). The preceding spring (March to April) NAO indicates
significant impacts on the following summer EASM in the interannual timescale (Wu
et al., 2009). Moreover, the signals originating from the atmosphere (e.g., NAO, EASM,
EAWM) and ocean (e.g., ENSO, PDO) present strong seasonality, prevailing in
different seasons. As shown by the fact that AC over China are impacted by various
climate systems, the relative importance of individual signals on their possible impacts
in modulating the variability of AC remains unknown. Meanwhile, most of the previous
studies regarding the influence of climate systems on AC focused on a certain season
with little attention paid to spatial-temporal variability. These questions are important
for improving the recognition of the modulation of climate systems on AC.
Consequently, one of the crucial motivations of the current work is to investigate
the spatial-temporal variability in the monthly AC over China, highlighting the potential
effects of climatic variabilities in modulating the spatial and temporal variations in AC,
and understanding the possible physical processes involved. The rest of the study is
arranged as follows. The model, datasets, and methods are presented in Section 2; The





properties of the leading modes of AC variability are described in Section 3; Section 4
discusses the contribution of climatic modes on aerosol variabilities; and Section 5
provides the conclusions and discussions.

## 2.  Datasets, model, and methodology

### 2.1  Model

The GEOS-Chem model is employed to detect the variability in AC over China.

This model is a 3-dimensional tropospheric chemistry model with a 2.5° longitude × 2°
latitude horizontal resolution and 30 vertical levels. The model is widely applied to
investigate the potential modulation of climatic variabilities on the anomalous
distributions of pollutants on various timescales, for example, on the seasonal
(Generoso et al., 2008; Jeong et al., 2011; Feng et al., 2019), interannual (Jeong et al.,
2017; Li et al., 2019), and interdecadal (Zhu et al., 2012) timescales. The high
consistency in both the temporal and spatial distributions between the simulations and
observations provides confidence for the feasibility of the present study.

As reported, the significant upward trend in anthropogenic emissions over China

accounts for a large variance in pollutants, and the first dominant mode of boreal winter
aerosols over eastern China represents anthropogenic emissions (Zhao et al., 2016). To
highlight the modulation of the climatic variabilities on the variation in the aerosols,
the anthropogenic and biomass burning emissions have been fixed at the year 2005
level. Thus, the variations in the aerosols in this context are attributed to the internal
climatic variability.
The definition of particulate matter smaller than 2.5 μm in diameter (PM2.5) is
followed by Liao et al. (2007),
$$[PM_{2.5}] = 1.29 \times [NO_3^-] + 1.37 \times [SO_4^{2-}] + [SOA] + [POA] + [BC]$$
where $NO_3^-$, $SO_4^{2-}$, SOA, POA, and BC are the aerosol particles of nitrate, sulfate,
secondary organic aerosol, primary organic aerosol, and black carbon, respectively.
Mineral dust and sea salt are excluded because these species are not the major
components over China.
**2.2  Datasets and methodology**
The input meteorological variables of the model highly agree with the widely used
atmospheric and oceanic datasets, i.e., the National Centers for Environmental
Prediction/National Center for Atmospheric Research (NCEP/NCAR) reanalysis
(Kalnay et al., 1996), and the UK Meteorological Office Hadley Centre's sea ice and
sea surface temperature (SST) datasets (HadISST; Rayner et al., 2003). These two
datasets are employed to verify the climatic indices calculated based on the model input
meteorological datasets. ENSO was characterized by the Niño 3.4 index, which is
defined as the areal averaged SST over 120°W-170°W, 5°N-5°S. The monthly Niño 3.4
indices based on the HadISST and model input data are highly related with each other
with a correlation coefficient of 0.99, confirming the reliability of the model data. The
North Atlantic Oscillation index (NAOI) and Northern Hemisphere Annular Mode
index (NAMI) are used to present the sea level pressure (SLP) oscillation between the
mid-latitudes and high latitudes in the extratropical Northern Hemisphere. Following



Li and Wang (2003), the NAMI is defined as the difference in the normalized global
zonal-mean SLP between 35°N and 65°N, in which the 35°N and 65°N refer to the mid-
latitude and high latitude, respectively. The definition of the NAOI resembles that of
the NAMI but within the North Atlantic sector from 80°W to 30°E. Because the NAOI
and NAMI are highly correlated with each other in both spatial distribution and
temporal variation (Thompson and Wallace, 1998; Gong et al., 2001), the NAOI is
utilized in the current context; similar results are obtained based on the NAMI.
Empirical orthogonal function (EOF) analysis was employed to obtain the
spatiotemporal variability in monthly $PM_{2.5}$ over China. Correlation and regression are
used to display the linkages between the variability in the $PM_{2.5}$ and the climatic modes.
Here, the period 1986–2006 was taken as the climatological mean, and the annual cycle
was removed before the analyses.

## 3. Distinct leading modes of the variability in aerosol concentrations

### 3.1 Two leading modes

Figure 1 presents the spatial distribution of the first (EOF1) and second (EOF2)
leading modes based on the monthly surface layer and column AC anomalies. A similar
spatial distribution is observed in both the surface and column AC. The EOF1 and EOF2
modes explain 31.4% (37.0%) and 16.3% (14.1%) of the total variances for the surface
layer (column) AC, respectively. Based on the *North*'s rule, the two dominant modes
could be significantly separated from each other and from the rest of the eigenvectors
based on the analysis of the eigenvalues in the light of sampling error above the 0.05





significance level. The rest of the modes are not discussed for their relative less
explained variance or could not be well separated. The EOF1 mode displays a mono-
sign pattern, with the maximum located in central eastern China (Figs. 1a and c). The
EOF2 mode presents a meridional dipole pattern in eastern China, with opposite values
to the south (positive values) and north (negative values) of the Yangtze River.
The temporal behavior of the two modes, the first and second principal
components, i.e., PC1 and PC2, is displayed in Figure 2. Both PC1 and PC2 show strong
interannual variations. The PCs based on the surface and column concentrations are
closely correlated with each other, with coefficients of 0.80 and 0.79 for PC1 and PC2,
respectively. The high consistency between the surface and column concentrations in
both the spatial and temporal distributions implies that the factors governing their
variations are the same. The maximum value of PC1 occurs in 1998, corresponding to
the strongest El Niño event (1997/98) in the 20th century. For PC2, negative values are
observed during the winters of 1989 and 2002, and positive values are observed during
the winters of 1995 and 1997. However, the winters of 1989 and 2002 correspond to
the positive polarities of the NAM or NAO, and the winters of 1995 and 1997 are
paralleling to the negative polarities of the NAM or NAO. The potential linkage
between the PCs and climatic variabilities is therefore analyzed. Here, the Niño 3.4
index is utilized to depict the variation of ENSO, and the NAOI (NAMI) is employed
to reflect the variability in the NAO (NAM). Note that the indices based on the model
input data are highly correlated with the observation datasets, and the monthly NAOI
is closely related with the NAMI, exhibiting a significant correlation coefficient of 0.71



during period 1986-2006. Therefore, the NAOI is employed to detect the linkage
between the PC2 and climate variability.
**3.2  Linkage with the climate variabilities**

Figure 3 displays the lead-lag correlation between the PC1 and Niño 3.4 index,

and between the PC2 and NAOI to identify the linkage between the climatic
variabilities and the two leading AC patterns. PC1 is significantly connected with the
Niño 3.4 index, with the maximum occurring when the Niño 3.4 index is 3 months
leading, implying a leading influence on PC1. The leading impacts of Niño 3.4 on the
variation in PC1 are further seen from the seasonal evolution of the standard deviation
in the corresponding indices (Fig. 4). The standard deviation of the monthly Niño 3.4
index shows that the maximum occurs during December, while the maximum occurs in
March for that of PC1. The leading influences of Niño 3.4 on PC1 are further verified
by the spatial distribution of correlations between PC1 and SST, as shown in Figure 5.
For the correlation with the PC1 lagged for 3 months, significant positive correlations
are observed over the tropical eastern Pacific and Indian Oceans, and negative
correlations over the tropical western Pacific. The correlation pattern is like a canonical
El Niño pattern. Note that the significant positive correlations over the tropical eastern
Pacific gradually decrease as the SST leading time is reduced; however, the correlations
over the tropical Indian Ocean become stronger, implying the effects of the Indian
Ocean capacitor along with the development of an ENSO event (Xie et al., 2009). The
above result ascertains the preceding influence of ENSO on the variation in PC1,
indicating a 3-month leading impact of ENSO on the following AC over China.


Meanwhile, the maximum negative correlation between PC2 and the NAO is
simultaneous (Fig. 3b), implying a simultaneous impact of the NAO on the AC over
China. Similar result is seen in the correlation between the NAMI and PC2. The
simultaneous relationship between the PC2 and NAO is further estimated in their
corresponding seasonal variation in the standard deviation (Figs. 4b and d). The
maximum standard deviations of the NAO and PC2 both occur during January-
February-March. A similar result is obtained based on the NAMI, suggesting significant
negative impacts of the extratropical atmosphere variation on the AC over eastern China.
Moreover, the correlations between the simultaneous PC2 and SLP display a negative
NAO-like (NAM-like) structure, with significant positive correlations over the polar
regions and negative correlations over the mid-latitudes. Note that this anomalous
pattern is consistently observed in PC2s based on both the surface layer and the column
concentrations.
The result above suggests that the variability in AC can be measured by climatic
variabilities, of which the variation in EOF1 is linked to the 3-month leading SST
variation over the tropical eastern Pacific, and that of EOF2 is related to the
extratropical atmospheric variability-NAO. The possible physical process involved in
their relationship is discussed in the following section.
**4.   Physical processes impacting on the leading modes**
**4.1  Circulation anomalies associated with ENSO**



Figure 7 shows the anomalous circulations associated with ENSO to identify the
atmospheric circulation process impacting the EOF1 patterns with the Niño 3.4 index
leading for 3 months. It is seen that tropical eastern Pacific and southern China are
controlled by significant positive correlations in the correlation with the divergence in
the lower troposphere. That is, southern China and tropical eastern Pacific are
influenced by anomalous convergence circulation under the influence of a 3-month
leading ENSO signal. Meanwhile, tropical western Pacific is impacted by significant
negative correlations, indicating that these regions are impacted by anomalous
divergence. The anomalous convergence circulation over southern China is not
favorable for the emission of AC. That is the anomalous circulation associated with 3-
month leading ENSO signal would connect with enhanced AC over eastern China,
which agrees with the spatial distribution of EOF1. Moreover, the impacts of ENSO on
the circulation is further seen in impacting the PBLH (Figure 8a). Significant negative
correlations are found over eastern China, indicating that the occurrence of a warm
ENSO event would decrease the height of PBLH. The decreased PBLH relates to
enhanced AC over eastern China. The above result suggests that the leading ENSO
signal exhibits a significant role in affecting the circulation anomalies over China.
Under the influence of warm ENSO events, the followed anomalous convergence and
decreased PBLH over eastern China are both unfavorable for the emission of AC,
contributing to the formation of the EOF1 pattern.
**4.2  Circulation anomalies associated with NAO**



The anomalous divergence accompanied by the simultaneous NAO is presented in
Figure 9. The northern Atlantic Ocean is influenced by an anomalous tripole structure,
showing convergence-divergence-convergence anomalies from the polar region to the
tropical regions. The occurrence of the anomalous circulation structure in the northern
Atlantic Ocean is due to the fact that the variation in NAO would induce an anomalous
tripole SST pattern within the northern Atlantic Ocean (e.g., Wu et al., 2009; Zheng et
al., 2016) by which a downstream wave-train is expected to be observed (Ruan et al.,
2015; Li and Ruan, 2018). The downstream wave train is seen with significant positive
anomalies over southern China in the regression of NAOI to the divergence, while
negative anomalies occur over northern China. That is, a positive NAO is accompanied
with anomalous divergence (convergence) over southern (northern) China. The
anomalous convergence over northern China is unfavorable for the emission of AC,
corresponding to enhanced AC. However, the opposite situation is observed over
southern China. The anomalous circulation connected with NAO further estimates the
negative impacts of NAO on the EOF2 mode.
In addition, the potential impacts of NAO on PBLH over China are further
examined. Figure 8b shows the anomalous PBLH regressed with reference to the NAOI
to identify the role of NAO in determining the EOF2 mode. For a positive NAO phase,
negative PBLH anomalies occupy northern China, suggesting a favorable condition for
enhanced AC. In contrast, southern China is controlled by positive PBLH anomalies,
paralleling the situation for decreased AC. The circulation anomalies connected with
NAO in both the divergence and PBLH suggest that the impacts of NAO on the AC



over northern and southern China are opposite, consistent with the spatial distribution
of the EOF2 mode.
**4.3 Role of temperature**
Meanwhile, it has been reported that temperature shows an effect in impacting the
distribution of aerosols. For example, it is reported that an increase in temperature is
associated with a decrease in $PM_{2.5}$ over southern California (Aw and Kleeman, 2003)
because enhanced temperature lead to decreases in organics and nitrate (Dawson et al.,
2007). Accordingly, the associated impacts of the ENSO and NAO on the temperature
over China are detected. Figure 10 displays the anomalous temperature regressed
against the 3 months preceding Niño 3.4 index and simultaneous NAOI to detect the
temperature anomalies connected with the two climate systems. For a warm event of
ENSO, large areas of negative temperature anomalies occupy eastern China, with the
maximum lying within the Yellow River and Yangtze River (Fig. 10a). The negative
temperature anomalies imply a lower temperature condition, which would induce to
enhanced AC.
For the NAO, its positive phase corresponds to opposite temperature anomalies
over southern and northern China, being positive (negative) over the southern (northern)
China (Fig. 10b). Positive temperature anomalies over southern China parallels to a
warmer situations and reduced AC in this region. Negative temperature anomalies over
northern China set up a background of colder situations, which would increase the AC.
The anomalous variation in the temperature agrees with the negative impact of NAO



on the AC over eastern China. In addition, the temperature anomalies accompanied with
the preceding ENSO are greater than those associated with the simultaneous NAO,
highlighting the dominant role of ENSO in impacting the AC over eastern China.
**5.  Conclusions and Discussions**
China has a high loading of aerosols and understanding the variability in AC is
important not only for recognizing the interactions between aerosols and climate but
also for scientifically understanding the current pollutant status. In the present work, it
is shown that the month-to-month AC over China are dominated by two principal
modes: the mono-pole mode and the meridional dipole mode. The first mono-pole mode
mainly exhibits the enhanced AC pattern over eastern China. The dipole mode shows
two centers over northern and southern China, with positive (negative) values over
southern (northern) China. The potential linkages between the two modes and climatic
sources are further described. The first mono-pole mode is linked with the 3 months
preceding ENSO, and the second dipole mode is connected with the simultaneous NAO.
The possible physical mechanism is also investigated by examining the dynamic
and thermal processes involved. For the mono-pole mode, the preceding ENSO can
induce anomalous convergence and decrease PBLH over eastern China, which are not
favorable for the emission of AC. Meanwhile, it is seen that anomalous negative
temperature over eastern China are seen accompanied with the preceding ENSO events,
paralleling conditions favorable for enhanced AC. For the meridional dipole mode,
anomalous convergence (divergence) and decreased (increased) PBLH are found over





northern (southern) China, paralleling the conditions for increased (decreased) AC
under the positive phase of NAO. Moreover, the temperature anomalies associated with
the NAO over southern and northern China are opposite, agreeing well with the spatial
distribution of the dipole mode. That is, both the dynamic and thermal anomalies
associated with climate systems are contributed to formation of the leading variabilities
of AC over China.
On the other hand, as reported, wet deposition shows important effects in
influencing the anomalous distribution of AC (Wu, 2014). However, the role of wet
deposition is not discussed in the present work. This is because the influences of ENSO
on the seasonal rainfall over China is complex and vary along with the phases of ENSO
events. During the decaying summer of a warm ENSO event, above average rainfall is
expected to be observed over southern China (e.g., Huang and Wu, 1989; Feng et al.,
2016); however, this is not the case for the developing summer (Feng et al., 2016).
Moreover, when the intensities of the ENSO events are different, i.e., moderate events
vs. strong events, their impacts on the seasonal rainfall over China may vary differently
(Xue and Liu, 2008). In addition, it has been indicated that the influence of rainfall on
the aerosols exhibits seasonal and regional dependence (Wu, 2014), and it is found that
the role of rainfall is limited in affecting the winter aerosols over southern China (Wu,
2014). However, the month-to-month variability of AC is considered in this study,
whereas for a specific season, the potential impacts of wet deposits in determining the
distribution of aerosols is complex and uncertainties exist.
Furthermore, the characteristics of the month-to-month variability of aerosols over



China is explored, the result highlights the impacts of tropical SST (i.e., ENSO) and the
atmospheric system (i.e., NAO or NAM) originating from the Northern Hemisphere on
the variability in AC over China. As reported, both ENSO and NAO display
considerable influences on the climate anomalies over China (e.g., Huang and Wu, 1989;
Zhang et al., 1996; Gong and Wang, 2003; Li and Wang, 2003), and the result here
expands their influences beyond climate. Climate systems, for example, originating
from the Southern Hemisphere, display essential influences in affecting seasonal
rainfall and temperature anomalies via atmospheric bridges and oceanic bridges (Zheng
et al., 2015, 2018). Future work will further examine the potential impacts of the
Southern Hemisphere climate systems on the variation in AC over China to
comprehensively assess the modulations of climate systems on the AC over China.





***Author contribution***

JLZ and JF conducted the study design. JLZ performed the simulations. JF and
JLZ carried out the data analysis. JPL and HL were involved in the scientific
interpretation. JF prepared the manuscript with contributions from all coauthors.

***Data availability***

The HadISST dataset are downloaded from
http://www.metoffice.gov.uk/hadobs/hadisst/data/download.html. The NCEP/NCAR
reanalyses is downloaded from http://www.esrl.noaa.gov/psd/data/gridded/. Simulation
results and codes to generate figures in this paper have been archived by corresponding
authors and are available at https://doi.org/10.5281/zenodo.3247326.

***Acknowledgements***

This research has been supported by the National Natural Science Foundation of
China (grant nos. 41790474, 41705131, and 41975079).



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



**Figure Captions:**

**Figure 1**. Spatial pattern of the (a) first empirical orthogonal function (EOF1) mode of the monthly surface PM2.5 concentrations over China. (b) As in (a), but for the second mode (EOF2). (c)-(d) As in (a)-(b), but for the column concentrations. The numbers indicate fractional variance in the EOF modes.

**Figure 2**. (a) The first principal components (PC1) of the monthly PM2.5 concentrations where the red and blue lines are for the surface and column concentrations, respectively. (b) As in (a), but for PC2.

**Figure 3**. (a) Lead-lag correlation between the Niño3.4 index and PC1. Negative (positive) lags indicate that the Niño3.4 index is leading (lagging) and the dashed lines are the 0.05 significance levels. (b) As in (a), but for the correlation between the NAOI and PC2. The red lines are based on the GEOS-4 meteorological fields, and the blue lines are based on the observations.

**Figure 4**. Seasonal variations in the standard deviation of the (a) PC1, (b) PC2, (c) Niño3.4 index, and (d) NAOI.

**Figure 5**. Spatial distribution of the correlation coefficients between the monthly sea surface temperature and PC1 for PC1 lagging for (a) 3 months, (b) 2 months, (c) 1 month, and (d) simultaneous. Color shading indicates significance at the 0.05 level.

**Figure 6**. Spatial distribution of the correlation coefficients between the monthly sea level pressure and PC2 of the (a) surface and (b) column PM2.5 concentrations. Color shading indicates significance at the 0.05 level.





**Figure 7**. Spatial distribution of the correlation coefficients between the Niño3.4 index
and divergence at 700 hPa for the Niño3.4 index leading for 3 months. Color
shading indicates significance at the 0.05 level.
**Figure 8**. Regressions of the divergence at 300 hPa onto the simultaneous NAOI. Color
shading indicates significance at the 0.05 level.
**Figure 9**. Regressions of the planetary boundary layer height onto the (a) 3-month
leading Niño3.4 index and (b) simultaneous NAOI. Color shading indicates
significance at the 0.05 level.
**Figure 10**. Regressions of the skin temperature onto the (a) 3-month leading Niño3.4
index and (b) simultaneous NAOI. Color shading indicates significance at the 0.05
level.





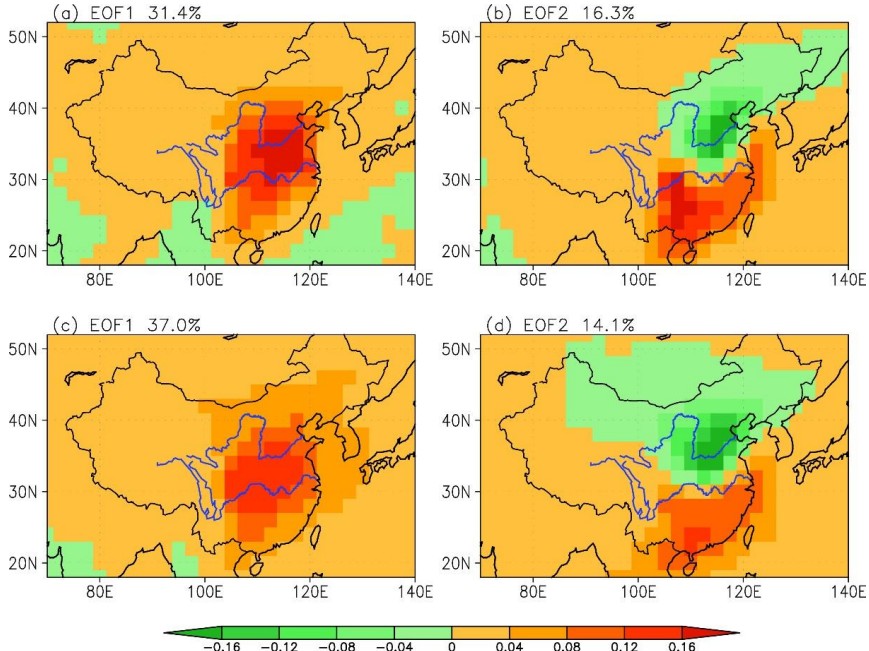

**Figure 1**. Spatial pattern of the (a) first empirical orthogonal function (EOF1) mode of the monthly surface PM2.5 concentrations over China. (b) As in (a), but for the second mode (EOF2). (c)-(d) As in (a)-(b), but for the column concentrations. The numbers indicate fractional variance in the EOF modes.



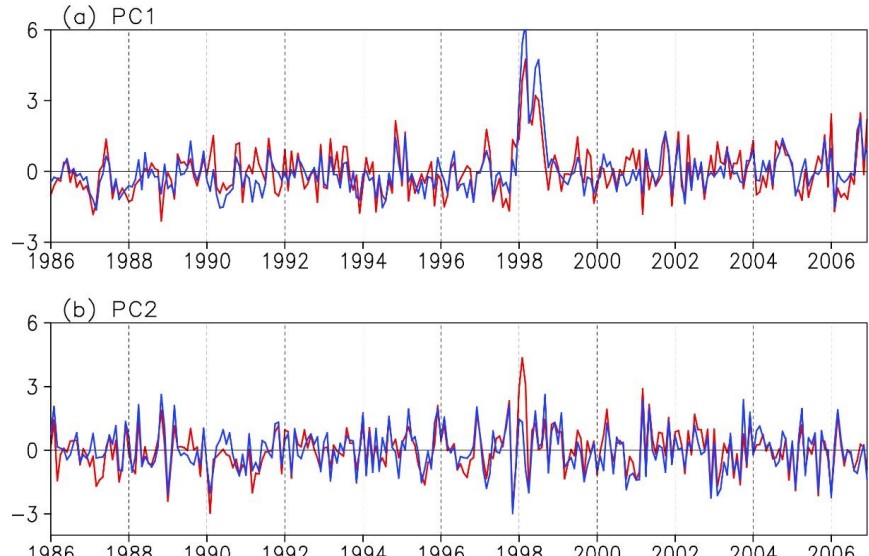

582

**Figure 2**. (a) The first principal components (PC1) of the monthly PM2.5
concentrations where the red and blue lines are for the surface and column
concentrations, respectively. (b) As in (a), but for PC2.

586

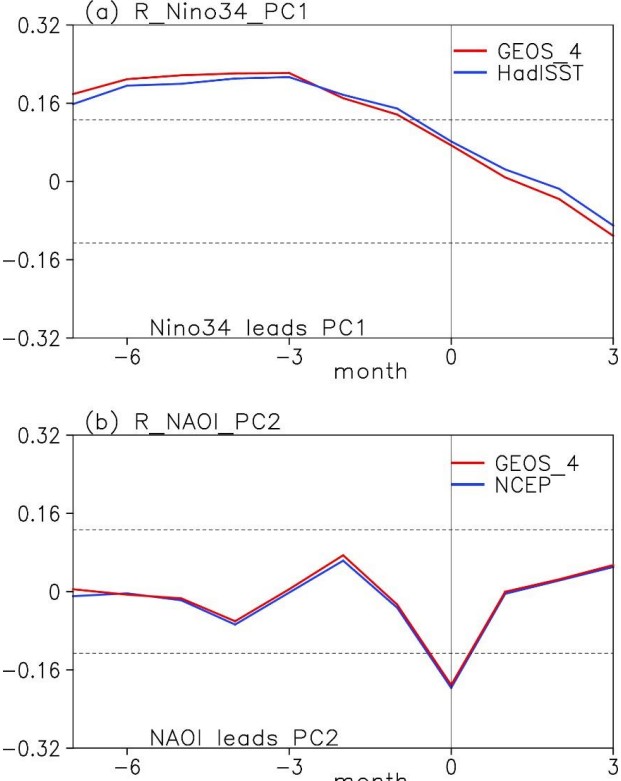

587

**Figure 3**. (a) Lead-lag correlation between the Niño3.4 index and PC1. Negative
(positive) lags indicate that the Niño3.4 index is leading (lagging) and the dashed lines
are the 0.05 significance levels. (b) As in (a), but for the correlation between the NAOI
and PC2. The red lines are based on the GEOS-4 meteorological fields, and the blue
lines are based on the observations.






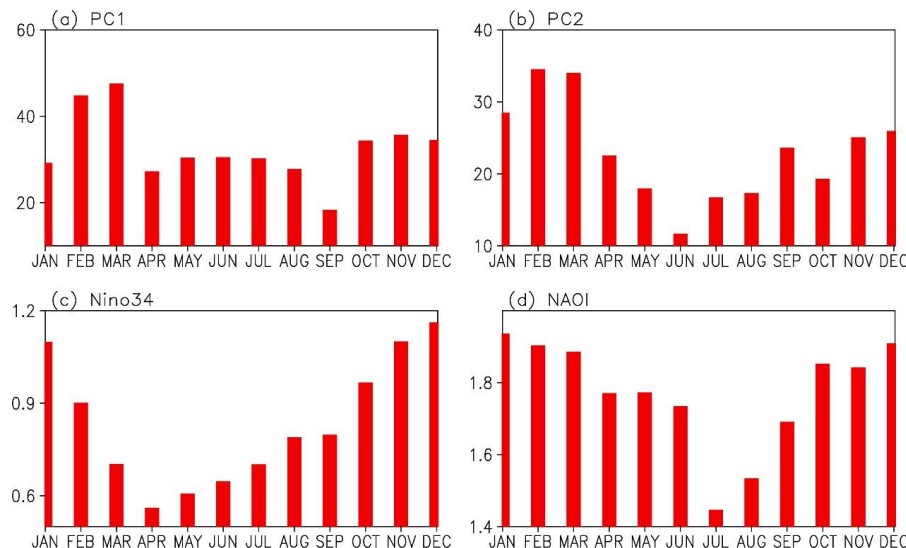


**Figure 4**. Seasonal variations in the standard deviation of the (a) PC1, (b) PC2, (c)

Niño3.4 index, and (d) NAOI.

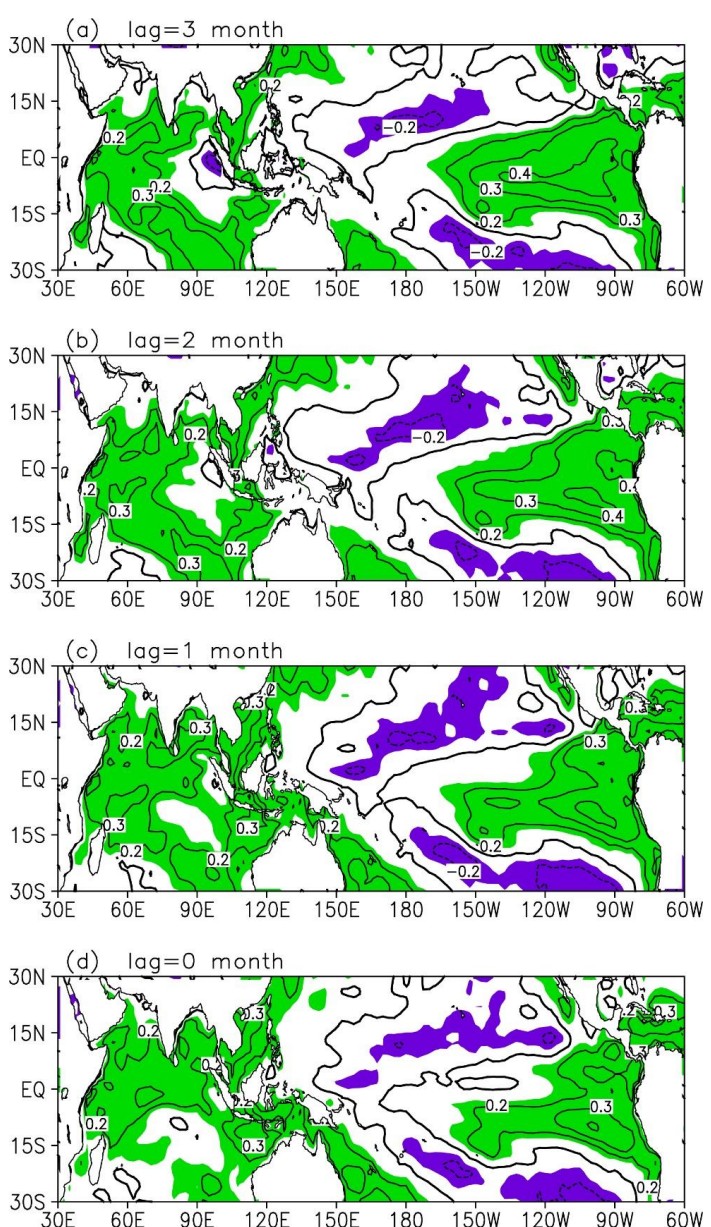


**Figure 5**. Spatial distribution of the correlation coefficients between the monthly sea
surface temperature and PC1 for PC1 lagging for (a) 3 months, (b) 2 months, (c) 1
month, and (d) simultaneous. Color shading indicates significance at the 0.05 level.

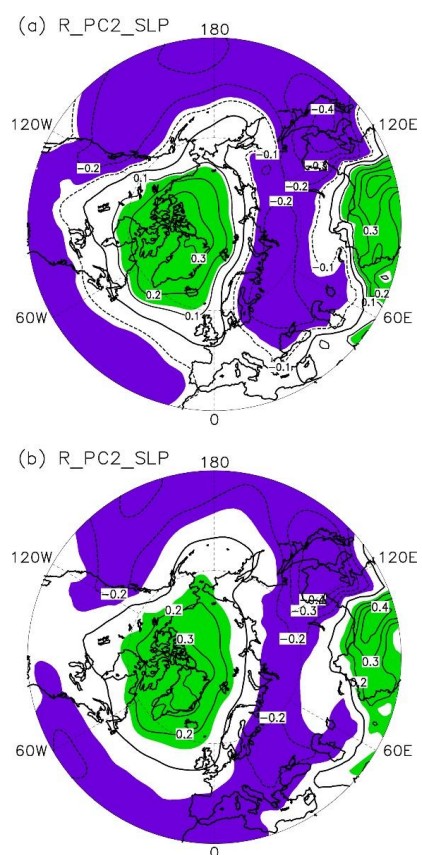


**Figure 6**. Spatial distribution of the correlation coefficients between the monthly sea

level pressure and PC2 of the (a) surface and (b) column PM2.5 concentrations. Color

shading indicates significance at the 0.05 level.






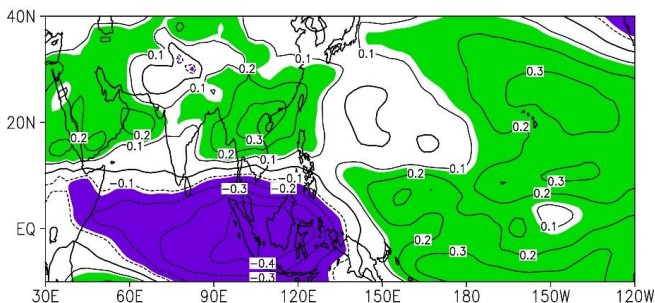


**Figure 7**. Spatial distribution of the correlation coefficients between the Niño3.4 index

and divergence at 700 hPa for the Niño3.4 index leading for 3 months. Color shading

indicates significance at the 0.05 level.




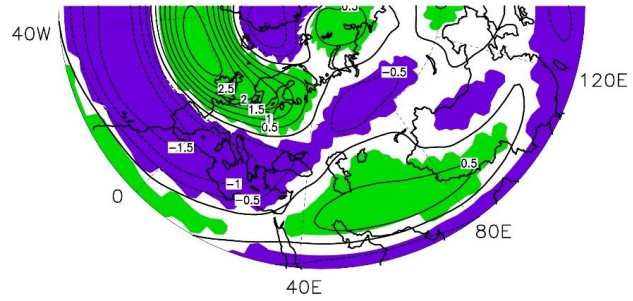


**Figure 8**. Regressions of the divergence at 300 hPa onto the simultaneous NAOI. Color

shading indicates significance at the 0.05 level.


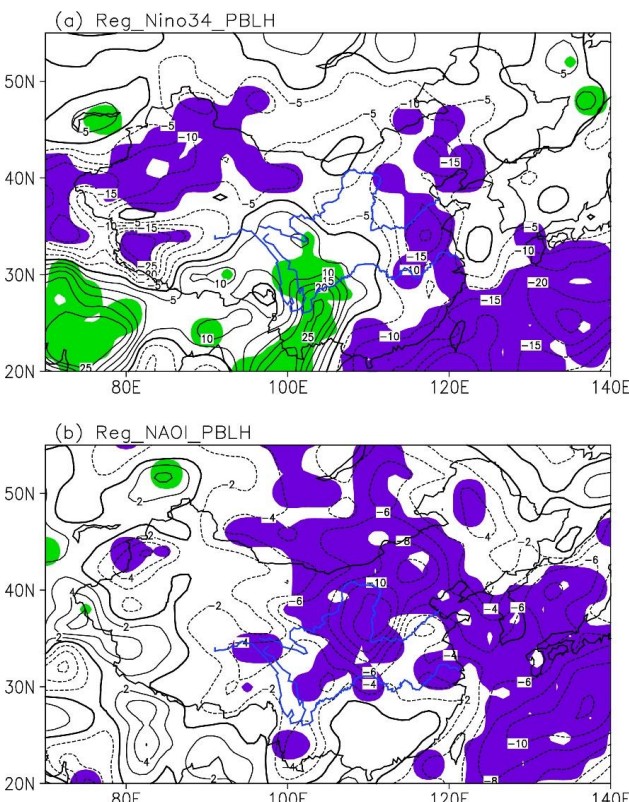

**Figure 9**. Regressions of the planetary boundary layer height onto the (a) 3-month leading Niño3.4 index and (b) simultaneous NAOI. Color shading indicates significance at the 0.05 level.

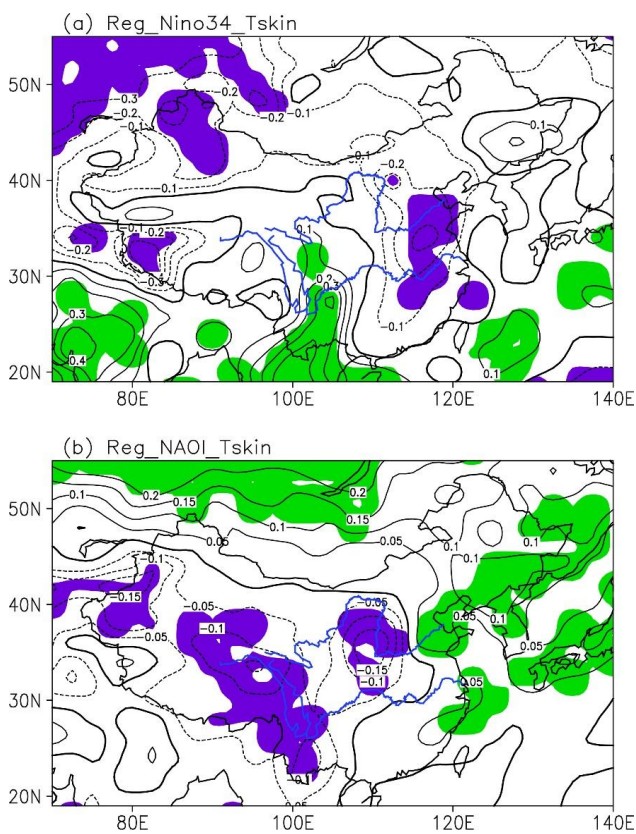

622

**Figure 10**. Regressions of the skin temperature onto the (a) 3-month leading Niño3.4

index and (b) simultaneous NAOI. Color shading indicates significance at the 0.05 level.