# Peer review of "Aerosol concentrations variability over China"

_Atmospheric Chemistry and Physics, 2019_

## Referee Comment (RC1) · Anonymous Referee #1 · 1 Apr 2020

General Comments

This paper studied the month-to-month variability of aerosol concentrations (AC) over China using a GESO-Chem model. The emission level in the model is set to a constant level of the year 2005. They found that two distinct lead modes dominate the natural variability: one is monopole mode which is related to the 3-month leading ENSO, while the second one is meridional dipole mode which is related to the NAO. The underlying physical mechanism is further analyzed. The results show that dynamical stability associated with the change of low-level convergence and planetary boundary layer height and thermal condition both play important roles.

Overall, the paper is well written and easy to understand. The topic is perfectly in line with ACP journal. Therefore, I recommend publishing after a minor revision.

[Figure]

Specific Comments

1. Title: Since this paper focus on the internal climatic variability, the emission level is fixed at the year 2005. Otherwise, the first leading mode might show an increasing trend due to the dominate role of anthropogenic emissions according to the previous study. Therefore, I suggest a title changed to: Aerosol concentrations natural variability over China: two distinct leading modes

2. Line 247 and other places: "emission" is not approximate here. How about using "transmission"?

3. Line 309: month-to-month variability of AC

---

## Referee Comment (RC2) · Anonymous Referee #2 · 29 Apr 2020

This paper tries to identify the climatic contribution to monthly aerosol variability over China using the GEOS-Chem model and tele-connection methodology. Though generally well written, this reviewer finds that more work needs to be done before it can be published. The major concerns include 1) model evaluation – there is a limited meteorological evaluation (against NCEP reanalysis and Hadley SST) but no evaluation of aerosol simulation at all, which makes audiences hard to gauge how meaningful the result is; 2) emissions – the authors fixed the emissions at the 2005 level to single out the climatic effect. However, aerosol has profound effects on climate through aerosol-cloud-radiation interactions. Different level of aerosol should have different feedback in the climate system. Ideally, the authors should conduct two experiments with high and low emissions to draw the conclusion. If that is not possible, the authors should at least discuss this point in the manuscript; 3) a few statements in the text were not consistent with the corresponding figures; and 4) driving meteorology – it appears GEOS-4 meteorology has been utilized to drive GEOS-Chem, which is very outdated. Why was the GEOS-5 not used?

Other specific comments include:

1. Lines 116 – 124: More details are needed for model description, e.g., driving meteorology, emissions inventories, etc. Were the biogenic emissions included in the simulation? If so, was it online calculated? Meteorology plays an important role in regulating biogenic emissions. A 2 x 2.5 degree model resolution appears very coarse for aerosol simulation. Were the results expected to change if a higher resolution simulation were conducted and analyzed?
2. Line 137: mineral dust is an important contributor to aerosol loading at least in spring over the northern China. Would exclusion of dust skew the results?
3. It is not clear why the authors correlated PC1 to Nino 3.4 index while correlated PC2 to NAOI. The explanation in lines 185 ~ 197 was not convincing. Have the alternative correlations been tried?
4. Figure 3. Is the y-axis correlation coefficient? Add this information on the plots.
5. In Lines 226-227, add something like as shown in Figure 6 to reference that figure.
6. Lines 238-242: Does the positive correlation of 3-month leading Nino 3.4 index and 700 hPa divergence mean anomalous convergence circulation?
7. Lines 246 – 247 and 270 (and in Conclusion): Should it be "not favorable for the emission of aerosol"? Are there any reference(s) for this statement?
8. Line 250: should it be Figure 9a?
9. Line 260: should it be Figure 8?
10. Line 275: should it be Figure 9b? (please make figure number consistent in the text and figure section).
11. Line 278: It appears that the positive PBLH anomalies are not significant based on Figure 9b.
12. Lines 296 – 298: Figure 10b does not support this statement.

---

## Author Comment (AC1) · 10 Jun 2020

The comment was uploaded in the form of a supplement:
https://www.atmos-chem-phys-discuss.net/acp-2019-1194/acp-2019-1194-AC1-supplement.pdf

---

## Author Comment (AC2) · 10 Jun 2020

**Response to Comments of Reviewer A**

**Manuscript number**: acp-2019-1194

**Author(s)**: Juan Feng, Jianlei Zhu, Jianping Li, and Hong Liao

**Title**: Aerosol concentrations variability over China: two distinct leading modes

**Response to Reviewer A**

**Overview comment:**

*This paper studied the month-to-month variability of aerosol concentrations (AC) over China using a GESO-Chem model. The emission level in the model is set to a constant level of the year 2005. They found that two distinct lead modes dominate the natural variability: one is monopole mode which is related to the 3-month leading ENSO, while the second one is meridional dipole mode which is related to the NAO. The underlying physical mechanism is further analyzed. The results show that dynamical stability associated with the change of low-level convergence and planetary boundary layer height and thermal condition both play important roles.*

*Overall, the paper is well written and easy to understand. The topic is perfectly in line with ACP journal. Therefore, I recommend publishing after a minor revision.*

**Response to general comment:**

Thanks to the reviewer for the helpful comments and suggestions. We have revised the manuscript seriously and carefully according to the reviewer's comments and suggestions. More details could be found in the revised manuscript.

**Specific comment:**

1. *Title: Since this paper focus on the internal climatic variability, the emission level is fixed at the year 2005. Otherwise, the first leading mode might show an increasing trend due to the dominate role of anthropogenic emissions according to the previous study. Therefore, I suggest a title changed to: Aerosol concentrations natural variability over China: two distinct leading modes.*

**Response:**

Thanks. We agree with the reviewer's comment that the natural variability of the aerosol concentrations is discussed in the present work, however, the aerosol concentrations is mainly the anthropogenic emissions. In addition, the natural aerosols, for example, mineral dust is not included in the manuscript. To avoid the misunderstanding, we have not changed the title.

2. *Line 247 and other places: "emission" is not approximate here. How about using "transmission"?*

**Response:**

This has been revised.

3. *Line 309: month-to-month variability of AC*

**Response:**

This has been revised.

**Response to Comments of Reviewer B**

**Manuscript number**: acp-2019-1194

**Author(s)**: Juan Feng, Jianlei Zhu, Jianping Li, and Hong Liao

**Title**: Aerosol concentrations variability over China: two distinct leading modes

**Response to Reviewer B**

**Overview comment:**

*This paper tries to identify the climatic contribution to monthly aerosol variability over China using the GEOS-Chem model and tele-connection methodology. Though generally well written, this reviewer finds that more work needs to be done before it can be published. The major concerns include 1) model evaluation – there is a limited meteorological evaluation (against NCEP reanalysis and Hadley SST) but no evaluation of aerosol simulation at all, which makes audiences hard to gauge how meaningful the result is; 2) emissions – the authors fixed the emissions at the 2005 level to single out the climatic effect. However, aerosol has profound effects on climate through aerosol-cloud-radiation interactions. Different level of aerosol should have different feedback in the climate system. Ideally, the authors should conduct two experiments with high and low emissions to draw the conclusion. If that is not possible, the authors should at least discuss this point in the manuscript; 3) a few statements in the text were not consistent with the corresponding figures; and 4) driving meteorology – it appears GEOS-4 meteorology has been utilized to drive GEOS-Chem, which is very outdated. Why was the GEOS-5 not used?*

**Response to general comment:**

Thanks to the reviewer for the helpful comments and suggestions. We have revised the manuscript seriously and carefully according to the reviewer's comments and suggestions. We would like to clarify the above comments in the following points,

**1. As to the comment** "*model evaluation – there is a limited meteorological evaluation (against NCEP reanalysis and Hadley SST) but no evaluation of aerosol simulation at all, which makes audiences hard to gauge how meaningful the result is*"

1) **As to the reliability of the meteorology fields**: we have shown in the manuscript, the input surface skin temperature of GEOS-Chem is highly correlated with the widely used SST dataset HadISST, in both spatial distribution and the long-term variability. Moreover, the input meteorological fields, such as winds, temperature, humidity, have been evaluated in Zhu et al. (2012a, 2012b) and Feng et al. (2016, 2017, 2019), and the result suggested the GEOS-Chem input meteorological fields are highly consistent with the NCEP/NCAR reanalyses. The above result provides confidence for the reliability of the meteorological fields of GEOS-Chem model.

2) **As to the reliability of the model simulations**:

a) Because of lacking valid aerosol concentrations observational data, we have not shown the comparison between the observations and simulations. However, previous studies have reported that the GEOS-Chem could well capture the seasonal and interannual variations of aerosol concentrations and $O_3$ over eastern China (including southern China; e.g., table 1 & fig. 1 in *Zhang et al.*, 2010; fig. 2 in *Wang et al.*, 2011; table 4 in *Lou et al.*, 2014; fig. 2 in *Yang et al.*, 2014)). In this study, we focused on the relative influences of climatic event on the aerosol concentrations instead of the absolute values of aerosol concentrations.

b) We have adopted the reviewer's comment by further comparing the spatial distribution of the aerosol concentrations based on the model simulations and MODIS AOD data (Figure R1). It is seen that similar spatial distributions are observed, e.g., the maximum is located to the north of Yangtze River during January, and to the north of Huanghe during July. That is both the seasonal evolution and spatial distribution of the aerosol concentrations over China is well reproduced in the GEOS-Chem.

c) We have compared the absolute values of aerosol species (i.e., Nitrate) based on the simulations and observations from other works (Table R1). It is seen that the simulated values of Nitrate in China based on the simulations and observations are equivalent.

The above discussions provide confidence for employing the GEOS-Chem to explore the influences of climatic events on aerosol concentrations, and it is proved to be a useful tool to understand the impacts of climatic event on aerosol concentrations without enough observations.

[Figure]

**Figure R1**. (Upper) The simulated aerosol concentrations over China during (left) January in 2001 and (right) July in 2001. (Below) As in the upper, but based on the MODIS AOD.

**Table R1**. The Nitrate concentrations based on the simulations and observations.

| Location | Value ($\mu g \ m^{-3}$) | | | | Reference |
|---|---|---|---|---|---|
| | Summer | | Winter | | |
| | Simulation | Observed | Simulation | Observed | |
| Beijing (121.3°E,31.1°N) | 9.82 | 11.18±10.37 | 10.71 | 12.29±12.12 | Wang et al.,2007 |
| Shanghai (121.3°E,31.1°N) | 7.07 | 4.76 | 17.95 | 10.10 | Ye et al., 2003 |
| Xi'an (108.6°E,34.2°N) | 3.43 | 14 | 13.23 | 56 | Zhang et al., 2002 |
| Nanjing (118.5°E, 32°N) | 12.64 | 3.24 | 37.22 | 12.9 | Yang et al., 2005 |
| Fuzhou (119.3°E,26.1°N) | 1.35 | 1.10±0.35 | 10.64 | 8.77±3.17 | Xu et al. 2011 |
| Hong Kong (114°E, 22.5°N) | 0.89 | 0.85±0.60 | 6.48 | 2.65±2.33 | Louie et al., 2005 |

**2. As to the comment** "*emissions …… Ideally, the authors should conduct two experiments with high and low emissions to draw the conclusion. If that is not possible, the authors should at least discuss this point in the manuscript*",

Thanks for the comment. The reviewer is right that aerosol has profound effects on climate through aerosol-cloud-radiation interactions, we have adopted the reviewer's comment by fixing the emissions at the level of year 1986 (low emission situation) and year 2006 (equivalent level of year 2005) to compare the role of different emissions on the distribution of aerosol concentrations. Four experiments are designed, meteorology field fixed on 1986 (M86), emission level at 1986 (Inv86) and at 2006 (Inv06), and meteorology field fixed on 2006 (M06), emission level at 1986 (Inv86) and at 2006 (Inv06). It is seen that aerosol concentrations significantly increased when the emission level increased, and the aerosol concentrations show a similar increasing trend under different meteorological conditions when the emission increases (Table R2). Note that the annual $SO_2$ emissions in eastern China increased from 4.54 Tg S yr$^{-1}$ in 1986 to 12.01 Tg S yr$^{-1}$ in 2006, with an increase rate of 164.5%, which is paralleling to the increase rate of $SO_4^{2-}$. Similar increase is seen in other particles.

On the other hand, it is seen that although the emission level is same, e.g., Inv06 but in M86 and M06, Inv86 but in M86 and M06, the output aerosol concentrations are different, suggesting that the role of meteorological conditions in impacting the aerosol concentrations. This point confirms the important modulation of the meteorological factors on the distribution of aerosol concentrations.

The above discussions indicate the role of meteorological conditions plays important role in impacting the distribution of aerosol concentrations. However, when the anthropogenic emissions have times increased, the variation of aerosol concentrations is mainly attributed to the emissions.

We have included the discussions into the revised manuscript for a better presentation.

**Table R2**. The differences of aerosol concentrations under different anthropocentric emissions averaged over eastern China (110°–125°E, 20°–45°N) during boreal winter.

| EX | M86 | | | | | M06 | | | |
|---|---|---|---|---|---|---|---|---|---|
| | concentrations (μg·m$^{-3}$) | | | Percentage | | concentrations (·μg·m$^{-3}$) | | | Percentage |
| species | Inv06 | Inv86 | Difference | (%) | | Inv06 | Inv86 | Difference | (%) |
| SO$_4^{2-}$ | 6.42 | 3.19 | 3.23 | 101.13 | | 7.07 | 3.63 | 3.44 | 94.75 |
| NO$_3^-$ | 13.88 | 5.90 | 7.98 | 135.35 | | 15.43 | 6.74 | 8.69 | 128.87 |
| NH$_4^+$ | 6.40 | 2.90 | 3.50 | 120.85 | | 7.09 | 3.3 | 3.79 | 114.9 |
| BC | 1.16 | 1.16 | <0.01 | <0.02 | | 1.24 | 1.24 | <0.01 | <0.01 |
| OC | 1.71 | 1.70 | <0.01 | <0.30 | | 1.8 | 1.8 | <0.01 | <0.13 |
| PM$_{2.5}$ | 29.56 | 14.84 | 14.72 | 99.19 | | 32.64 | 16.71 | 15.92 | 95.28 |

**3. As to the comment** "a few statements in the text were not consistent with the corresponding figures",

We carefully checked the manuscript and all the figure captions in the revised manuscript.

**4. As to the comment** *"driving meteorology – it appears GEOS-4 meteorology has been utilized to drive GEOS-Chem, which is very outdated. Why was the GEOS-5 not used?",*

We agree with the reviewer that the GEOS-4 is an old version of the model. The GEOS-5 is a relative new version; however, it is only available for the period 2004-2013. We have explored the month-to-month variability of the aerosol concentrations in this study, ten years data is not enough for a climatological analysis. Moreover, as mentioned above, the GEOS-4 is highly consistent with the widely used reanalysis dataset, which provide the reliability of the present work.

**Major comment:**

1. *Lines 116 – 124: More details are needed for model description, e.g., driving meteorology, emissions inventories, etc. Were the biogenic emissions included in the simulation? If so, was it online calculated? Meteorology plays an important role in regulating biogenic emissions. A 2 x 2.5 degree model resolution appears very coarse for aerosol simulation. Were the results expected to change if a higher resolution simulation were conducted and analyzed?*

**Response:**

Thanks for the comment. The biogenic emissions are included in the simulation and it is online calculated. As to the resolution of the model simulations, as mentioned this model is widely employed to explore the aerosol concentrations variations all over the world, including East Asia, and it is proved to be a useful and reliable tool to investigate the variation of the aerosol concentrations. Due to the compute capability and time limitation, we have not updated the resolution of the simulation, however, we will adopt the reviewer's comment and expect to update the simulation to a higher resolution to further examine the influence of meteorological conditions on the aerosol concentrations.

2. *Line 137: mineral dust is an important contributor to aerosol loading at least in spring over the northern China. Would exclusion of dust skew the results?*

**Response:**

The reviewer is right, mineral dust is an important contributor to the boreal spring aerosol concentrations over northern China. However, the month-to-month variability is explored in this study, rather than the spring. That is the time scale are different. In addition, mineral dust is mainly a kind of natural aerosols, however, we focused mainly the anthropogenic aerosols in this study.

3. *It is not clear why the authors correlated PC1 to Nino 3.4 index while correlated PC2 to NAOI. The explanation in lines 185 ~ 197 was not convincing. Have the alternative correlations been tried?*

**Response:**

We have performed the correlations between the PCs and other climatic indices, e.g., Indian Ocean Dipole index, ENSO Modoki index, NAO index, and Niño3.4 index. Significant relationships are observed between the PC1 and Niño3.4 index, and between the PC2 and NAO index. The correlation coefficients between the PC1 and NAO index is 0.09, and it is 0.05 between the 3-month leaded Niño3.4 index and PC2, both are insignificant. Therefore, only the relationships between the NAOI and PC2, and between the Niño3.4 index and PC1 are discussed in the manuscript.

4. *Figure 3. Is the y-axis correlation coefficient? Add this information on the plots.*

**Response:**

We have adopted the reviewer's comment and replotted the figure.

5. *In Lines 226-227, add something like as shown in Figure 6 to reference that figure.*

**Response:**

This has been revised.

6. *Lines 238-242: Does the positive correlation of 3-month leading Nino 3.4 index and 700 hPa divergence mean anomalous convergence circulation?*

**Response:**

Figure 7 shows the spatial distribution of the correlation coefficients between the Niño3.4 index and convergence for the Niño3.4 index leading for 3 months. The positive correlations over the tropical eastern Pacific and negative correlations over western Pacific are corresponding to anomalous Walker circulation.

7. *Lines 246 – 247 and 270 (and in Conclusion): Should it be "not favorable for the emission of aerosol"? Are there any reference(s) for this statement?*

**Response:**

This has been revised, we have included the relevant reference.

8. *Line 250: should it be Figure 9a?*

**Response:**

This has been revised

9. *Line 260: should it be Figure 8?*

**Response:**

This has been revised.

10. *Line 275: should it be Figure 9b? (please make figure number consistent in the text and figure section).*

**Response:**

Thanks, we have carefully examined the figure captions and figure numbers in the manuscript during the revision.

11. *Line 278: It appears that the positive PBLH anomalies are not significant based on Figure 9b.*

**Response:**

The two figures present different physical variables, i.e., Figure 9 is for the PBLH, and Figure 10 is for the temperature. The two figures both for the regression distribution, but based on different variables. The statistical significance of the regression values was evaluated by a two-sided Student's *t*-test.

*12. Lines 296 – 298: Figure 10b does not support this statement.*

**Response:**

Figure 10b shows the relationship between the NAOI and temperature. It is seen that positive phase corresponds to opposite temperature anomalies over southern and northern China. Southern China is mainly occupied with positive anomalies, and northern China is under influenced by negative anomalies in general. This result agrees with the relation between the NAO and AC. Moreover, we have carefully examined the whole manuscript to avoid clerical errors during the revision process. The detailed revisions are shown in the revised manuscript. We hope the revised manuscript could offset the shortcomings in the original manuscript.

[revised manuscript text omitted]